# Allergies to Allergens from Cats and Dogs: A Review and Update on Sources, Pathogenesis, and Strategies

**DOI:** 10.3390/ijms251910520

**Published:** 2024-09-29

**Authors:** Wei An, Ting Li, Xinya Tian, Xiaoxin Fu, Chunxiao Li, Zhenlong Wang, Jinquan Wang, Xiumin Wang

**Affiliations:** 1Institute of Feed Research, Chinese Academy of Agricultural Sciences, Beijing 100081, China; anweiyoux@163.com (W.A.); 82101232041@caas.cn (X.T.); fuxiaoxin2000@outlook.com (X.F.); 82101212039@caas.cn (C.L.); wangzhenlong02@caas.cn (Z.W.); 2Key Laboratory of Feed Biotechnology, Ministry of Agriculture and Rural Affairs, Beijing 100081, China; 3State Key Laboratory of Pathogen and Biosecurity, Beijing Institute of Biotechnology, No. 20, Dongda Street, Beijing 100071, China; liting7427@163.com

**Keywords:** allergy, pet allergen, epitope, IgY antibody, immunotherapy

## Abstract

Inhalation allergies caused by cats and dogs can lead to a range of discomforting symptoms, such as rhinitis and asthma, in humans. With the increasing popularity of and care provided to these companion animals, the allergens they produce pose a growing threat to susceptible patients’ health. Allergens from cats and dogs have emerged as significant risk factors for triggering asthma and allergic rhinitis worldwide; however, there remains a lack of systematic measures aimed at assisting individuals in recognizing and preventing allergies caused by these animals. This review provides comprehensive insights into the classification of cat and dog allergens, along with their pathogenic mechanisms. This study also discusses implementation strategies for prevention and control measures, including physical methods, gene-editing technology, and immunological approaches, as well as potential strategies for enhancing allergen immunotherapy combined with immunoinformatics. Finally, it presents future prospects for the prevention and treatment of human allergies caused by cats and dogs. This review will improve knowledge regarding allergies to cats and dogs while providing insights into potential targets for the development of next-generation treatments.

## 1. Introduction

Allergic reactions, also known as type I hypersensitivity, occur when the body’s pre-existing immune system is re-exposed to the same allergen, resulting in an excessive immune response mediated by immunoglobulin E (IgE). Common clinical symptoms include urticaria, eczema, asthma, rhinitis, vomiting, diarrhea, and potentially life-threatening anaphylactic shock [1]. This prevalent global condition involves ubiquitous allergens that bind to IgE antibodies and trigger the aforementioned reactions in daily life, such as pollen, food, dust mites, microorganisms, animal dander, and insect bites [2]. 

Among these allergens, cat and dog dander are considered significant factors in the development of allergic asthma and rhinitis, with their prevalence increasing over time [3]. Approximately 10–20% of surveyed individuals exhibit sensitivity to cat dander, while 12–18% are sensitive to dog dander [4,5]. In a study conducted in China to investigate allergies, it was found that both cats and dogs play a crucial role in triggering allergic reactions by assessing levels of allergen-specific antibodies in patients [6]. The growing population of cats and dogs in China implies an increasing threat posed by the presence of dog and cat allergens in the air. Furthermore, it has been observed that allergens from cats and dogs significantly contribute to allergic reactions worldwide (Table 1). Notably, among individuals with symptomatic sensitization in developed countries, the prevalence of sensitization to cats and dogs may exceed one-fifth [6]. Individuals who keep pets at home inadvertently transport allergens to their workplace, resulting in a 57.6% higher median concentration of allergen Fel d 1 in offices owned by cat owners compared to non-cat owners; similarly, there is a 77.14% higher median concentration of Can f 1 allergen in offices owned by dog owners compared to non-dog owners [7]. Even individuals lacking prior pet ownership experiences can detect the presence of cat and dog allergens within their households if they reside in communities with high rates of pet keeping [8]. Additionally, it has been observed that specific cat allergens persist in indoor air for a duration of 6–9 months even after pets have been removed [9]. These findings undoubtedly contribute to an increased risk of allergic diseases caused by pets, imposing significant health and economic burdens on the general public.

Currently, there is a lack of convenient and effective methods for the treatment of human allergies to cats and dogs. One predominant approach involves alleviating allergic symptoms through the administration of antihistamines, corticosteroids, or decongestants [23]. Another method is allergen immunotherapy, which needs gradual exposure to allergens to facilitate adaptation without eliciting excessive reactions. However, these treatments have prolonged durations, and desensitization vaccines are susceptible to adverse reactions [24].

In this study, we conducted a comprehensive literature survey on allergens from cats and dogs, providing detailed insights into their classification and pathogenic mechanisms. Moreover, we summarized recent advancements in strategies for the prevention and treatment of allergic diseases, followed by methods to enhance immunotherapy.

## 2. Sources of Allergens 

The World Health Organization (WHO) has registered eight allergens each for cats and dogs (Table 2). Among cat allergens, Fel d 1, Fel d 7, and Fel d 4 exhibit the highest frequencies of IgE recognition and the ability to activate basophilic granulocytes. The median allergen concentration required to reach the activation plateau in a basophilic leukemia cell line expressing human high-affinity IgE receptor (FcεRI) is approximately 0.1 ng/mL for Fel d 1, 1 ng/mL for Fel d 7, and also 1 ng/mL for Fel d 4; these levels are nearly 100-fold higher than those of Fel d 6 and Fel d 8 [25]. Among dog allergens, Can f 1 accounts for approximately 70% of allergies, while *Canis familiaris* prostatic kallikrein (Can f 5) and Can f 6 account for over 30% [26]. 

It should also be noted that individuals may have simultaneous sensitivity to multiple allergens due to cross-reactivity between cat and dog allergens caused by sequence conservation with a cross-sensitivity rate of 48% [44,45]. 

### 2.1. Lipocalin 

The allergens Fel d 4, Fel d 7, Can f 1, Can f 2, Can f 4, and Can f 6 from cats and dogs belong to the lipocalin family and commonly induce respiratory allergies. Lipocalins exhibit various functions, including retinol transportation, pheromone transport, prostaglandin synthesis, immune regulation, and cellular synthesis [46]. They consist of approximately 150–250 amino acids each with low conservation and homology. Dog Can f 1 only shares 25% similarity with human lipid transport protein 1 (LCN1) [47]. However, they also possess one to three structurally conserved regions (SCR1-SCR3) and exhibit a high degree of similarity in the tertiary structure. This structure is characterized by eight reverse parallel β-strands that form a central β-barrel closely associated with the α-helix. Within this barrel-like structure, hydrophobic ligand-binding sites are present, enabling them to carry various lipophilic small molecules and perform diverse physiological functions [31,48]. Furthermore, lipocalins possess the functional characteristic of acting as specific ligands that bind to cell surface receptors. Cat Fel d 4 can serve as a ligand activating the vomeronasal organ of mice, which is a specialized chemosensory epithelium responsible for eliciting defense responses against predators [49].

The allergen database contains a total of 19 lipid transport protein allergens from 10 mammalian species. However, our current knowledge on the mechanisms responsible for their capacity to elicit allergic reactions remains limited. Due to the tendency of lipid carrier proteins to form oligomers such as o dimers (Can f 4) or tetramers (Can f 2), these particular allergenic oligomers have the potential to affect FcεR I receptor cross-linking and subsequently exert an influence on mast cell activation [50]. The allergenicity of lipocalins may be associated with their capacity to bind specific cell surface receptors. Pierre et al. demonstrated the suppression of mannose receptor expression on dendritic cells (DCs) from human peripheral blood monocytes by gene silencing, thereby inhibiting the recognition and uptake of Can f 1 protein [51]. Additionally, other studies have indicated that cell surface heparan sulfate proteoglycans (HSPG) act as endocytic receptors involved in heterologous Fel d 4 uptake [52].

Due to their structural similarity, lipocalins produced by different species may possess identical antigenic determinants that can bind to the same antibodies, thereby exhibiting immune cross-reactivity between Can f 2, Can f 6, and Fel d 4. Regrettably, there is currently insufficient evidence regarding the correlation between the allergenicity of the lipid transport protein family and its structure and function. 

### 2.2. Immunoglobulin

Immunoglobulins are dimeric proteins consisting of two heavy and light chains, secreted by plasma cells derived from B cells. The immunoglobulins expressed on the surface of B cells are commonly referred to as B-cell receptors, while the soluble form of immunoglobulins is known as antibodies. Based on their structural characteristics, distribution patterns, composition, and functions, immunoglobulins include immunoglobulin G (IgG), immunoglobulin A (IgA), immunoglobulin M (IgM), immunoglobulin D (IgD), and IgE. Cat Fel d 5 and Fel d 6 correspond to IgA and IgM, respectively. The glycan portion of cats’ IgA serves as the primary antigen-binding site for human-produced IgE [3,53].

### 2.3. Serum Albumin 

Serum albumin is widely distributed across various mammalian species and belongs to a class of multifunctional proteins that exhibit high conservation in both sequence and structure. Its primary functions include the regulation of osmotic pressure, transportation of small molecules, and pH buffering [54]. Additionally, serum albumin has been identified as an allergen in humans. Cat Fel d 2 and dog Can f 3 are members of the serum albumin family; they exhibit immunological cross-reactivity not only with each other but also with serum albumin from other mammals such as horses [55]. This cross-reactivity can be attributed to the extensive preservation of antigenic epitopes resulting from sequence and structural similarity [56].

### 2.4. Glycoprotein

Secretoglobin Fel d 1, a representative feline allergen, is the primary causative factor responsible for human allergies. It exists as a tetrameric glycoprotein composed of two non-covalently linked heterodimers, each with an approximate molecular weight of 18 kDa [57]. Chain 1 of Fel d 1 shares approximately 30% sequence homology with other uteroglobin proteins, while chain 2 exhibits minimal homology [58]. Previous studies have demonstrated that males secret higher levels of Fel d 1 than females [59]. However, the precise mechanisms underlying its allergenicity and physiological functions remain unclear. It has been shown that Fel d 1 can significantly enhance immune signal transduction through Toll-like receptors (TLR4 and TLR2) on immune cell surfaces and can bind to lipopolysaccharide (LPS) to amplify TLR signaling. Furthermore, it has been found that the cell surface mannose receptor plays an essential role in DCs’ uptake of Fel d 1 by binding allergens via its cysteine domain to sulfated acidic polysaccharide structures on allergens [60]. 

However, upon comparing natural Fel d 1 with recombinant expressed Fel d 1 proteins exhibiting varying degrees of glycosylation in baculovirus and *Pichia pastoris* expression systems, a significant increase in tumor necrosis factor α (TNF-α) expression was observed in mouse bone marrow-derived macrophages when co-combined with LPS. Importantly, this enhancement was not elicited in TLR4-deficient mice. These findings suggest that the mechanism underlying Fel d 1’s activity may be related to ligand–protein interactions and the TLR4/TLR2-mediated immune pathway, which operates independently of the mannose receptor on DCs’ surfaces. Given that Fel d 1 is primarily located in glandular regions such as the cat’s perianal area, it could potentially function as a feline pheromone or serve as a carrier for pheromone transmission due to its ability to bind small molecule ligands within its hydrophobic cavity [61]. 

### 2.5. Cystatin

The proteins cat Fel d 3 and dog Can f 8 are membranes of the cystatin family, a class of natural reversible inhibitors that tightly bind to cysteine proteases. Given the widespread presence of cysteine proteases in organisms, which perform various physiological functions, cystatins play a crucial role in regulating these enzymes [55].

### 2.6. Esterase

The kallikrein-like prostatic allergen Can f 5, also known as arginine esterase, is a 28 kDa protein present in the urine and fur of dogs [62]. It is secreted by the prostate gland and under the regulation of male hormones and exclusively detectable in male dogs. Notably, it has been identified as the sole allergen in approximately one-third of individuals with dog allergies [38,63]. There have been reported cases where individuals who previously had no adverse reactions to semen exposure developed allergic symptoms upon exposure to dogs in public places [64]. Specific IgE against Can f 5 has been found to be significantly present. Furthermore, Can f 5 shares around 55–60% structural homology with human prostate-specific antigens and exhibits immunological cross-reactivity between them [65].

## 3. Pathogenic Mechanisms of Allergens

Cat and dog allergens are typically small molecular proteins synthesized in various glands or the liver. Once secreted, they primarily localize in the pet’s skin, hair, and body fluids such as saliva, milk, urine, and sweat. During shedding of hair and dander by animals, these allergens can adhere to dust particles and disperse throughout the environment. Upon inhalation of these allergens, antigen-presenting cells (APCs), and predominantly DCs, can capture them and present them to T-cell receptors (TCRs) through major histocompatibility complex (MHC) molecules on naive T cells’ surface. This process promotes their differentiation into helper 2 cells (Th2). Subsequently, Th2 cells secrete cytokines including interleukin 4 (IL-4), interleukin 5 (IL-5), interleukin 13 (IL-13), interleukin 9 (IL-9), and interleukin 10 (IL-10) that stimulate B lymphocytes to produce specific IgE antibodies targeting allergens while inducing their proliferation as well as differentiation into memory B cells or plasma cells. The plasma cells in the bone marrow can continuously produce IgE, which subsequently enters the systemic circulation and binds to FcεRI receptors on mast cells and basophils, thereby causing a transient asymptomatic sensitization state. Upon re-exposure to allergens, cross-linking of specific IgE molecules bound to mast cells or basophils triggers a signaling cascade leading to mast cell activation and degranulation (Figure 1). This process results in the release of pre-stored and newly synthesized mediators such as histamine, arachidonic acid, prostaglandins, causing urticaria, eczema, asthma, rhinitis, vomiting, and diarrhea [44,66,67,68,69,70].

A few studies have revealed that micronutrients, particularly iron, play a crucial role in regulating the immune system. Iron indirectly influences the proliferation, differentiation, and function of immune cells such as macrophages, mast cells, T cells, and B cells through various mechanisms involving iron regulatory factors and transporters [72]. Insufficient iron levels can activate antigen-presenting cells, promote Th2 cell generation, and facilitate class switching in B cells. Therefore, it has been hypothesized that inadequate iron content within immune cells may contribute to allergic reactions [73,74,75].

## 4. Strategies for Preventing and Treating Allergies

### 4.1. Physical Methods

The removal of pets from the household has been demonstrated as an effective approach in reducing adverse reactions among patients with pet allergies [76]. By conducting a comparative analysis of allergen content in 50 households with cats and 50 without, it was revealed that the discrepancy in allergen levels present in carpets and soft furnishings reached magnitudes as high as 200–300 times [77]. After removing cats from 15 households, there was a significant decrease in Fel d 1 level in the air by several hundred times over a period ranging from 9 to 43 weeks. Remarkably, one group even achieved allergen levels comparable to those observed in cat-free households [76]. Therefore, avoiding direct contact with pets remains the most straightforward and effective method for preventing human allergies.

Utilizing tap water or pet-specific bathing products for approximately 60 s can effectively reduce airborne allergens by over 40% [78]. Similarly, washing dogs at least twice a week can also lead to a significant reduction in airborne allergen concentration by at least 40% within a short period of time [79]. Due to the favorable solubility properties of most pet allergens, regular laundering of textiles using washing machines and appropriate detergents can eliminate over 95% of allergens [80]. Furthermore, air filters play an instrumental role in significantly reducing pet allergen levels in indoor environments [81,82]. By incorporating air filters, it is possible to achieve an impressive reduction rate of 84% for dog allergens within enclosed spaces (Figure 2) [83]. In another study, the proportion of patients with cat allergies who developed early asthma in a room containing cat allergens under placebo conditions was three times higher than those exposed to a room with cat allergens and an operational air purifier [84]. However, when Gore et al. tested the effectiveness of different brands of high-efficiency new vacuum cleaners and old control vacuum cleaners in removing allergens, they found that exposure to cat allergens increased by three to five times [85]. Although air purifiers can significantly reduce inhalable dust levels during household activities, there is no statistically significant reduction in pet allergen concentration [86]. Air purifiers can effectively decrease over 90% of indoor pet allergens and alleviate allergic symptoms in a mouse model. Furthermore, the photoelectrochemical oxidative (PECO) Molekule filtration device (PFD) demonstrates a significant reduction in eosinophil recruitment associated with allergies compared to the HEPA-assisted air filtration device (HFD). Further detailed experimental investigations are necessary to determine the efficacy of air filters in controlling pet allergen concentrations in different environments, using various brands of air filters, and considering varying numbers of pets present [87]. These physical methods for pet allergen removal, although simple and cost-effective, necessitate long-term persistence and pose challenges in achieving complete eradication.

### 4.2. Gene Editing

To effectively remove pet allergens, individuals are also exploring alternative long-term methods for reducing cat allergens. Gene-editing technology has been used to specifically target the genes responsible for producing allergen proteins at the genetic level, upstream of protein expression (Figure 2). The objective is to disable these genes and selectively breed cat breeds with reduced allergenicity [93]. By utilizing the CRISPR-Cas9 system and cytoplasm injection clone technology, gene-edited cats have been successfully generated with lower levels of Fel d 1 expression [88]. While this approach offers a permanent solution for eradicating pet allergens, there remains limited understanding regarding the precise function of Fel d 1, which raises uncertainty about potential adverse effects on both individuals and their offspring resulting from knocking out a single gene. Furthermore, considering the existence of at least eight identified cat allergens that can cause allergic reactions in humans, the genetic knockout of all these allergens would pose unknown risks and ethical challenges.

### 4.3. Immunological Strategies

#### 4.3.1. Allergen-Specific Immunotherapy

Allergen-specific immunotherapy (ASIT) is an effective method of desensitization for patients with IgE-related allergies (Figure 2). Continuous administration through subcutaneous immunotherapy (SCIT), sublingual immunotherapy (SLIT), and intralymphatic immunotherapy (ILIT) induces immune tolerance, thereby preventing the activation of an immune response against specific antigens [94,95]. Potential mechanisms of ASIT include a reduction in allergen-specific Th2 cells, activation of regulatory T and B cells, and production of “blocking” antibodies IgG and IgA [96]. In a three-year clinical trial involving 32 adults and children, cat and dog dander extracts were used for allergen-specific immunotherapy. The study observed a decrease in IgE levels, an increase in IgG levels, and a superior efficacy of cat immunotherapy compared to dog immunotherapy [89]. Unfortunately, 41% of patients with cat allergies who received subcutaneous injection of cat allergens experienced severe systemic reactions [97]. Therefore, ASIT is a promising desensitization method that requires further research on antigen preparations with high immunogenicity and minimal toxicity to ensure significant quality.

#### 4.3.2. Monoclonal Antibodies

Allergies are frequently triggered by the binding of allergens to IgE. A therapeutic approach involves identifying anti-IgE antibodies that can neutralize IgE and prevent its interaction with allergens. Akiko et al. developed monoclonal antibodies known as CRE-DR against dog IgE, which have been validated through enzyme-linked immunosorbent assay (ELISA) for their specific reactivity to human IgE, indicating the potential of these monoclonal antibodies in treating allergies [98]. Omalizumab is a humanized monoclonal antibody that selectively binds to circulating IgE and inhibits the activation of inflammatory cells and the release of inflammatory mediators [99]. It has successfully completed clinical trials for various types of allergic asthma and is commercially available in several countries. In a clinical study evaluating omalizumab for cat allergen-induced allergies, the group receiving omalizumab demonstrated a significant reduction in acute asthma, achieving an efficacy rate of 62.9% [100]. However, the administration of omalizumab may result in adverse reactions such as headaches and pharyngitis [101]. Currently, there is a lack of further clinical trials investigating the effectiveness of anti-IgE antibodies in alleviating allergies to cats and dogs. Therefore, future research should focus on exploring more effective and safer anti-IgE antibodies, as well as combination prevention and treatment strategies.

The synthesized human IgG4 antibodies REGN1908 and REGN1909 exhibit synergistic and competitive binding to allergens in a non-competitive manner, effectively inhibiting the interaction between allergens and IgE. This inhibition leads to the suppression of inflammatory mediator releases induced by allergens from eosinophils and mast cells. In vitro experiments have revealed that both antibodies exhibit a half-maximal inhibitory concentration (IC_50_) value of 0.45 nM, resulting in an 83% inhibition rate against IgE binding. Subsequent successful clinical trials conducted on mice and humans have demonstrated the alleviation of allergic symptoms [102].

The current availability of monoclonal antibodies in the market is limited due to their expensive production process and the previously mentioned adverse reactions, which pose challenges for widespread promotion.

#### 4.3.3. Immunoglobulin Y

Immunoglobulin Y (IgY) is a highly homologous polyclonal antibody produced by oviparous animals (such as poultry, reptiles, etc.), which specifically binds to corresponding antigens for antigen inactivation and plays a crucial role in passive immunity. Following immunizing poultry with specific antigens, the resulting antibodies are transported through the bloodstream to the ovaries, where they gradually accumulate within egg cells and ultimately store in the yolk. Each egg can yield approximately 50–100 mg of IgY, while an individual chicken can produce over 20 g of IgY per year. The advantages of IgY include a well-established preparation process, high production yield, short cycle time, and adherence to animal welfare principles [103].

Due to the distant genetic relationship between species, IgY exhibits an inherent high affinity for mammalian proteins [104]. Satyaraj et al. used specific chicken IgY targeting Fel d 1 to effectively inhibit the binding of Fel d 1 to IgE. In a feeding experiment conducted on cats, Fel d 1 levels in saliva were reduced by at least 20% when cats were fed with IgY antibodies [90]. Another study evaluated the safety of administering three different doses of anti-Fel 1 IgY antibodies to 27 kittens over a period of 84 d. The results showed that their weekly weight and physical condition scores remained within healthy ranges, and no abnormalities were observed during veterinary examinations and blood biochemistry analyses [105]. Undoubtedly, using specific IgY antibodies for neutralizing allergens produced by cats represents an economical, environmentally friendly, and safe approach; however, further cat feeding experiments and clinical evidence are still required for validation.

Additionally, β-lactoglobulin (BLG) is a well-known protein found in bovine milk whey. It binds to immune cells and carries iron, thereby activating the aryl hydrocarbon receptor (AHR) pathway for immune regulation and promoting resilience and tolerance among these cellular components [75]. Bergmann et al. developed lozenges containing holo-BLG-based micronutrients as supplements to prevent allergies to cats. Following a three-month treatment course with holo-BLG, 35 patients exhibited increased tolerance to allergen concentrations during nasal provocation tests compared to pre-treatment levels; 20 patients experienced no allergic late-phase reactions [106]. These findings highlight the potential efficacy of targeted micronutrients delivered through holo-BLG lozenges in preventing allergies to cats, as well as other allergens [91].

## 5. Strategies for Enhancing Allergen Immunotherapy

### 5.1. Vaccine Adjuvants and Modifications

The combination of different adjuvants with allergens can enhance the stability and immunogenicity of vaccines. Utilizing non-reactive immunological adjuvants in conjunction with antigens effectively stimulates relevant immune cells, thereby augmenting the immune response to the antigen [107]. Commonly employed adjuvants (such as aluminum hydroxide, alum, and Freund’s adjuvant) have demonstrated notable success in generating immunity against viral and bacterial infections; however, further improvements are still required to enhance the protective antibody response. Andersson et al. utilized carbohydrate-based particles (CBPs) covalently coupled with recombinant cat allergens, resulting in a significant increase (three to four times) in IL-8 release and TNF-α production by human monocyte-derived DCs [108]. Tasaniyananda et al. successfully employed liposomes as an immune adjuvant for specific immunotherapy in a murine model of feline allergy [109]. Similarly, Leonard et al. effectively treated the Fel d 1 allergic mouse model by using CpG oligodeoxynucleotides as an adjuvant [110]. Although there is growing development of novel adjuvants, there is still a lack of comprehensive experimental evaluations regarding their combination with pet allergens. In recent clinical trials, Corren et al. used tezepelumab, a human monoclonal antibody targeting thymic stromal lymphopoietin (TSLP), as an adjuvant for ASIT in patients with cat allergy. Tezepelumab reduced serum concentrations of IL-5 and IL-13, as well as total IgE levels in asthma patients while improving lung function and alleviating symptoms. Notably surpassing sole administration of SCIT, the incorporation of tezepelumab exhibited remarkable efficacy in reducing nasal reactions induced by allergens, with sustained effects even after one year [111].

Adjuvants not only effectively enhance the efficacy of ASIT but also reduce production costs by reducing the concentration and frequency of antigen administration. Different adjuvants exhibit distinct effects on antigens [112]. Currently, some novel adjuvants such as liposomes, oil-in-water emulsions, and TLR agonists have shown positive effects in ASIT. However, there are limited evaluations regarding the combined effects of adjuvants with diverse cat and dog allergens.

The safety of immune preparations can be enhanced by chemically modifying allergens to disrupt their IgE binding sites while preserving immunogenicity [113]. González et al. polymerized the natural extract of cat dander using glutaraldehyde, resulting in safer immune preparations that still maintain efficacy [114]. Calzada et al. modified Can f 1 and Can f 5 using glutaraldehyde and found that the polymer exhibited low IgE binding while effectively activating basophils in dog allergic patients [115]. However, it is yet to be proven whether common chemicals like formaldehyde, glutaraldehyde, and tyrosine are safe and effective for modifying pet allergens [116].

### 5.2. Recombinant Allergens

Extracting and purifying allergens from pets is undoubtedly an inefficient and time-consuming process, prompting scientists to focus on heterologous recombinant expression systems. Schmitz et al. successfully conjugated Fel d 1 with bacteriophage Qbeta-derived virus-like particles (Qbeta-Fel d 1), resulting in the fusion protein Qβ-Fel d 1. However, this highly immunogenic protein does not elicit basophil degranulation in humans with cat allergies or induce allergic reactions in mouse models [117]. Katarzyna et al. expressed a fusion protein in *Escherichia coli* by combining the PreS domain derived from hepatitis B virus with two non-allergenic peptides of Fel d 1. In mouse and rabbit immunization experiments, this fusion protein exhibited similar properties to Qbeta-Fel d 1 as it not only induced Fel d 1-specific IgG antibodies but also inhibited the binding of IgE from allergic patients to the allergen [118]. Ogrina et al. fused Fel d 1 with virus-like particles obtained from eggplant mosaic virus (EMV) and achieved high titers of Fel d 1-specific antibodies in mice [119]. By genetically fusing allergens with carriers that enhance stability and immunogenicity, fusion proteins exhibit higher immunogenicity while simultaneously reducing IgE reactivity. These fusion proteins can be efficiently produced through large-scale fermentation processes, thereby offering potential avenues for the development of ASIT vaccines.

### 5.3. Epitopes

Epitopes refer to specific regions on the surface of antigens that possess a distinct structure and immunological activity, capable of stimulating antibody production or sensitizing lymphocytes. With further research studies on the structure and sensitization mechanism of allergens, scientists are increasingly focusing on the antigenic determinants (epitopes) of these proteins (Table 3). These antibodies or lymphocytes recognize epitopes, which typically consist of less than 20 amino acid residues. Based on their continuity or discontinuity within the three-dimensional (3D) protein structure, epitopes can be classified as either linear or conformational [120].

Compared to intact allergens, artificially synthesized fusion epitopes not only encompass dominant protein epitopes but also include hidden epitopes located within the antigenic molecules [126]. Norman et al. conducted a single-blind subcutaneous injection treatment on 95 cat-sensitive patients over a four-week period using two synthesized peptides, each containing 27 amino acids and based on the T-cell epitopes of Fel d 1 (ALLERVAX CAT). The efficacy of the treatment was evaluated by nasal and pulmonary symptoms in patients exposed to rooms occupied by live cats before and after the intervention. Following the four-week treatment, both the middle- and high-dose groups demonstrated significantly improved tolerance to cats compared to the control and low-dose groups [127]. Worm et al. developed Cat Peptide Antigen Desensitization (Cat-PAD), which is a combination of seven equimolar peptides targeting Fel d 1 [128]. This epitope vaccine effectively alleviated allergic symptoms in patients with cat allergies without causing any difference in acute allergic reactions, fluctuations in vital signs, or adverse events when compared to a placebo group in a clinical trial involving 14 allergic children [129].

## 6. Prospects of Prevention of Allergies

Allergens from pets remain a crucial factor in the induction of allergic rhinitis in humans. Currently, there are limited effective approaches to prevent allergies caused by cats and dogs, including allergen elimination or immune reaction inhibition. However, there is a lack of comprehensive research on integrating diverse preventive methods. During allergen production stages, pets can be fed with IgY antibodies specifically targeting allergens, while regular utilization of air purifiers can effectively eradicate residual airborne allergens. Furthermore, coordinated immunological interventions involving micronutrients supplementation with holo-BLG can be implemented. By combining preventive measures that address distinct phases of the allergic response, it is possible to minimize allergy occurrence.

However, achieving complete eradication of environmental allergens remains unattainable, rendering individuals with cat allergies constantly susceptible and highlighting the imperfections in preventing human allergic reactions to cats and dogs. The IgY antibody treatment lacks comprehensive long-term clinical trial data, while additional allergens such as Cat-NPC2, Fel d S100, and Fel d Hp have been identified [130,131]. Current preventive strategies primarily target Fel d 1 and Can f 1 but fail to provide sufficient protection for the entire population. The primary challenge lies in facilitating precise allergen diagnosis for patients through accurate allergen screening methods, thereby offering personalized treatment strategies. Clinically, inhalant allergen screening methods predominantly consist of skin prick tests (SPTs), intradermal tests, allergen nasal provocation tests, and serum IgE assays [132]. Notably, a report of 622 cat allergy sufferers revealed that 27.8% of patients tested positive exclusively on SPT, while only 2.6% tested positive solely on specific IgE assays. To overcome the occurrence of false negatives, multiple testing methods are recommended [133].

Once the specific allergen is identified, desensitization treatment can be combined with ASIT. This involves formulating a vaccine containing purified allergens and specific immunological adjuvants, which are administered in gradually increasing concentrations to induce immune tolerance towards the allergen in patients. Previous studies have demonstrated that the efficacy of ASIT is influenced by various factors. One challenge lies in the isolation and purification of allergens from natural extracts, which directly affects the subsequent purity and cost of allergen reagents [134]. Additionally, there is a concern regarding potential adverse reactions associated with natural allergen extracts [135].

Fortunately, advancements in proteomics, immunomics, and bioinformatics can effectively mitigate the adverse reactions, reduce costs, and expedite development timeframes of ASIT. Databases such as Uniprot (http://www.uniprot.org/, accessed on 15 August 2024) provide researchers with comprehensive annotations for over 60 million proteins [136], while repositories like the RCSB Protein Data Bank (PDB) contain an extensive collection of over 210,000 3D structures [137]. When obtaining an antigen protein sequence through high-throughput sequencing, these databases can be easily utilized to identify sequences exhibiting the highest conservation and homology to the target antigen. Furthermore, computational models such as SWISS-MODEL and AlphaFold2 accurately predict the 3D structures of unknown antigens. This enables a deeper understanding of their function and facilitates structural characterization of antigenic epitopes [138,139]. Numerous machine learning-based algorithms have been developed for predicting antigenic epitopes based on known protein structures. These tools facilitate the prediction and screening of various lymphocyte-binding epitopes, enabling the subsequent fusion of selected epitopes into a protein construct [140]. Tools like Expasy (https://www.expasy.org/, accessed on 15 August 2024) and AlphaFold2 can analyze secondary and tertiary structures, while servers such as ClusPro 2.0 (https://cluspro.org/login.php, accessed on 15 August 2024) and C-ImmSim (http://150.146.2.1/C-IMMSIM/index.php, accessed on 15 August 2024) allow for molecular docking simulations and immune simulations to assess the viability of the fusion epitope construct. Immunoinformatics-based vaccine design has demonstrated successful effects with other allergens (Table 4). Moten et al. used a series of immunoinformatics techniques to develop a vaccine targeting Amb a 11, a ragweed pollen protein. Serum tests conducted on patients allergic to ragweed revealed that the predicted MHC II epitopes were capable of stimulating T cells to generate IgG antibodies against ragweed pollen without being detected by IgE antibodies [141]. Finally, genetic engineering or chemical methods are employed to synthesize proteins, followed by the selection of appropriate adjuvants; molecular biology methods can be used to verify the immunogenicity of the fusion epitope protein in vitro.

## 7. Conclusions

Allergies induced by cats and dogs continue to pose a significant threat to public health. Various allergens have been identified in these pets, and their pathogenic mechanisms have also been summarized. For individuals, implementing physical preventive measures (such as avoiding contact with pets in public places, utilizing indoor air purifiers, and practicing frequent clothes washing) and administering IgY antibodies to pets are cost-effective strategies. Although ASIT is the most clinically effective approach, it is crucial to carefully consider the economic burden and potential adverse reactions due to current technological limitations. Gene-editing techniques also offer the possibility of a permanent cure in the future. This review also discussed strategies for enhancing the effectiveness of immunotherapy and explored future prospects that can contribute to the prevention and control of human allergies to other allergens.

## Figures and Tables

**Figure 1 ijms-25-10520-f001:**
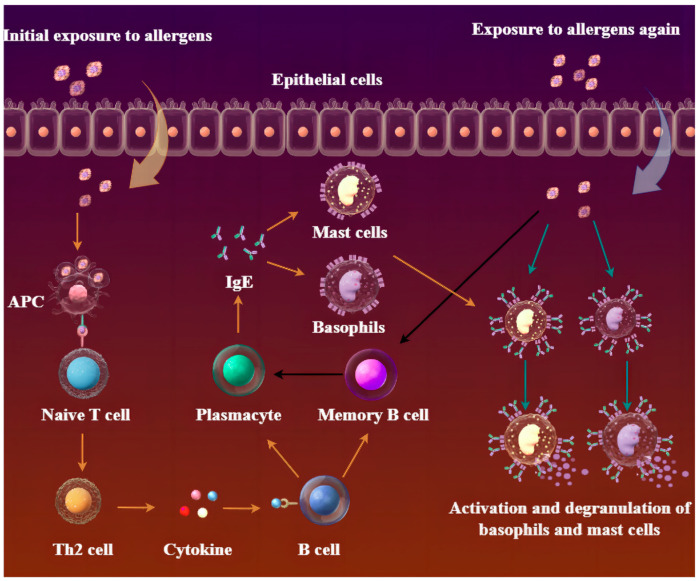
The process of allergic reactions. Antigen-presenting cells (APCs) process allergens that enter the body and further present them to T cells for activation (The yellow arrow represents the immune pathway following the first exposure to allergens; the black arrow represents the activation of memory B cells by the body in response to subsequent allergen exposure, leading to IgE production; the blue arrow represents the direct binding of IgE to allergens after the second exposure). Under the stimulation of specific cytokines, such as IL-4 and IL-13 secreted by Th2 helper T cells, B cells undergo proliferation and differentiation, leading to the formation of memory B cells. These B cells then produce and release specific IgE antibodies against allergens. Subsequently, these specific IgE antibodies bind to high-affinity IgE receptors on mast cells and eosinophil surfaces, inducing an allergenic state in the body without exhibiting corresponding symptoms. Upon re-exposure to the allergen, cross-linking occurs between two or more IgE Fc fragments on mast cells and eosinophil surfaces, causing degranulation of mast cells along with histidine release and subsequent occurrence of allergic symptoms [66,68,69,71].

**Figure 2 ijms-25-10520-f002:**
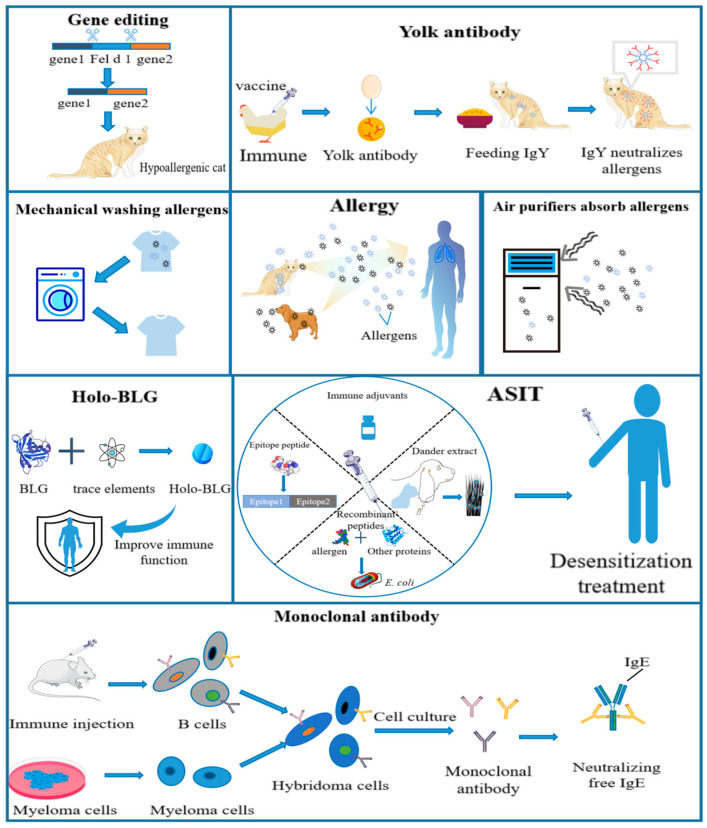
Methods for preventing human allergies to cats and dogs. A schematic diagram illustrates various methods for preventing human allergies caused by cats and dogs, including physical methods (such as mechanical washing, air purifier, etc.), a gene-editing technique, and immunological approaches (such as egg yolk antibody, holo β-lactoglobulin (holo-BLG), allergen-specific immunotherapy (ASIT), monoclonal antibody, etc.) [83,87,88,89,90,91,92].

**Table 1 ijms-25-10520-t001:** Frequency of sensitization to cat and dog allergens in different countries.

Countries	Methods	Total	Cats	Dogs	Reference
China	Blood testing	24,057	8.60%	6.10%	[10]
Russia	Blood testing	513	24.10%	21.40%	[11]
South Korea	Skin prick test	7504	20.60%	15.20%	[12]
Germany	Blood testing	356	34.80%	31.70%	[13]
Japan	Blood testing	12,205,097	18.20%	18.90%	[14]
America	Blood testing	478	54.40%	64.70%	[15]
Canada	Skin prick test	623	53.10%	17.30%	[16]
Qatar	Skin prick test	473	6.18%	0.50%	[17]
Lebanon	Skin prick test	919	29.90%	21.90%	[18]
Thailand	Skin prick test	1516	12.90%	10.00%	[19]
Nepal	Skin prick test	170	15.30%	14.10%	[20]
Mexico	Skin prick test	761	26.70%	33.90%	[21]
Egypt	Enzyme allegro sorbent test	122	4.10%	6.60%	[22]

**Table 2 ijms-25-10520-t002:** Allergenic components of cat and dog allergens.

Pets	Allergens	Chemical Essence	Molecular Weight (kDa)	Sources	Sensitization Capacity	Cross-Reaction	References
Cats	Fel d 1	Secretoglobin	38	Dander, saliva	84.2% of 361 patients	NA ^1^	[27]
	Fel d 2	Serum albumin	69	Blood, dander	11.9% of 361 patients	Can f 3, Sus s 1, Bos d 6, Equ c 3	[27,28]
	Fel d 3	Cystatin A	11	Dander, blood	50.6% of 36 patients	Can f 8	[29,30]
	Fel d 4	Lipocalin	22	Saliva	31% of 361 patients	Can f 6, Can f 2	[27,31]
	Fel d 5	Immunoglobulin A	400	Saliva, blood	NA	NA	[25]
	Fel d 6	Immunoglobulin M	800–1000	Saliva, blood	32.8% of 24 patients	NA	[25]
	Fel d 7	Lipocalin	17.5	Saliva	31.3% of 361 patients	Can f 1	[27]
	Fel d 8	Latherin-like protein	24	Saliva	42.4% of 32 patients	NA	[32]
Dogs	Can f 1	Lipocalin	23–25	Dander, saliva	69.6% of 23 patients	Fel d 7	[33,34]
	Can f 2	Lipocalin	19–27	Dander, saliva	25% of 20 patients	Fel d 4	[33,35]
	Can f 3	Serum albumin	69	Blood, sander	35% of 110 patients	Fel d 2, Equ c 3	[36]
	Can f 4	Lipocalin	16–18	Dander	76% of 37 patients	Cow dander extract	[37]
	Can f 5	Prostatic kallikrein	28	Urine, seminal plasma	76% of 37 patients	PSA	[37,38]
	Can f 6	Lipocalin	27–29	Dander	61% of 44 patients	Fel d 4, Equ c 1	[39,40]
	Can f 7	Niemann pick type C2 protein	16	Dander	10–12% of 71 patients	Cat NPC2	[41,42]
	Can f 8	Cystatin	14	Dander, blood	12% of 245 patients	Fel d 3	[43]

^1^ NA: not available.

**Table 3 ijms-25-10520-t003:** Possible antigenic epitopes of cat and dog allergens.

Sources	Allergens	Epitopes	References
Cats	Fel d 1	Chain 1: 1MKGACVLVLLWAALLL16, 60ARILKNCVDAK70, Chain 2: 1MRGALLVLALLVTQ14, 43DLSLTKVNATEPER56	[121]
	Fel d 1	Chain 1: L34, T37; T39; P40, E42, E45; R61, K64, N65, D68; E73, K76	[122]
	Fel d 1	Chain 1: 41DEYVEQVAQYKALPVVLENA60, 78NALSVLDKIYTSPLC92	[123]
	Fel d 2	51KALPVVLENARILKN65, 21AETCPIFYDVFFAVA35, 26IFYDVFFAVANGNEL40, 36NGNELLLDLSLTKVN50, 66YVENGLISRVLDGLV80	[124]
	Fel d 4	56VFVEHIKALDNSSLS70, 96YTVVYDGYNVFSIVE110, 116YILLHLLNFDKTRPF130	[125]
Dogs	Can f 1	His86, Arg152	[124]
	Can f 3	156QLFLGKYLYEIARRH170, 176QLFLGKYLYEIARRH190, 421KLGEYGFQNALLVRY435, 476FLSVVLNRLCVLHEK490, 531FTFHADLCTLPEAEK545	[123]
	Can f 6	30SKI32, 46KEK48, 50EEN52	[24]
	Can f 6	43SDIKEKIEENGS54, 76TKVNGKCT83, 91KTEKDGE97, 125NVNQEQEF132, 139GRKPDVSPKVKEKF152	[125]

**Table 4 ijms-25-10520-t004:** Antigen epitope prediction tools of cat and dog allergens.

Name	Epitope Type	Method or Algorithm	Website	References
EpiJen	T-cell epitope (MHC I)	Multi-step algorithm	https://www.ddg-pharmfac.net/epijen/EpiJen/EpiJen.htm (accessed on 15 August 2024)	[142]
BigMHC	T-cell epitope (MHC I)	Algorithm based on deep neural networks	https://github.com/KarchinLab/bigmhc (accessed on 15 August 2024)	[143]
NetMHCpan	T-cell epitope (MHC I)	Algorithm based on artificial neural networks	http://www.cbs.dtu.dk/services/NetMHC/ (accessed on 15 August 2024)	[144]
ProPred1	T-cell epitope (MHC I)	Algorithm based on matrix	NA	[142,145]
ProPred	T-cell epitope (MHC II)	Algorithm based on matrix	https://github.com/wjguan/propred/blob/master/README.md (accessed on 15 August 2024)	[146]
MHCPred	T-cell epitope (MHC I, II)	Algorithm based on partial least squares method	http://www.ddg-pharmfac.net/mhcpred/MHCPred/ (accessed on 15 August 2024)	[147]
Syfpeithi	T-cell epitope (MHC I, II)	The algorithms used are based on the book “MHC Ligands and Peptide Motifs”	http://www.syfpeithi.de/ (accessed on 15 August 2024)	[148]
NetmMHCIIpan	T-cell epitope (MHC II)	Algorithm based on artificial neural networks	https://services.healthtech.dtu.dk/services/NetMHCII-2.3/ (accessed on 15 August 2024)	[149]
TepiTool	T-cell epitope (MHC I, II)	Algorithm based on deep learning	http://tools.immuneepitope.org/tepitool/ (accessed on 15 August 2024)	[150]
Vaxign2	T-cell epitope (MHC I, II)	Algorithm based on reverse vaccinology and machine learning	https://violinet.org/vaxign2 (accessed on 15 August 2024)	[151]
NETCTLpan	T-cell epitope (MHC I)	Algorithm based on artificial neural networks and weight matrices	https://services.healthtech.dtu.dk/services/NetCTLpan-1.1/ (accessed on 15 August 2024)	[152]
SVMHC	T-cell epitope (MHC I, II)	Algorithm based on support vector machine	NA	[153]
ABCPred	B-cell epitope	Algorithm based on artificial neural network	https://github.com/chaninlab/abcpred (accessed on 15 August 2024)	[154]
BCPred	B-cell epitope	Algorithm based on the subsequence kernel	http://ailab-projects2.ist.psu.edu/bcpred/predict.html (accessed on 15 August 2024)	[155]
DiscoTope3.0	B-cell epitope	Algorithm based on AlphaFold2 modeling and inverse folding latent representations	https://services.healthtech.dtu.dk/services/DiscoTope-3.0/ (accessed on 15 August 2024)	[156]
EPSVR	B-cell epitope	Algorithm based on support vector regression	http://sysbio.unl.edu/EPSVR/ (accessed on 15 August 2024)	[157]
SEPPA3.0	B-cell epitope	Algorithm based on logistic regression model	http://www.badd-cao.net/seppa3/index.html (accessed on 15 August 2024)	[158]
BEpro	B-cell epitope	Algorithm based on combination of amino acid propensity scores and half-sphere exposure values at multiple distances	https://pepito.proteomics.ics.uci.edu/ (accessed on 15 August 2024)	[159]
COBEpro	B-cell epitope	Algorithm based on support vector machine	http://scratch.proteomics.ics.uci.edu/ (accessed on 15 August 2024)	[160]
ElliPro	B-cell epitope	Residue clustering algorithm	http://tools.iedb.org/ellipro/ (accessed on 15 August 2024)	[161]
ScanNet	B-cell epitope	Algorithm based on geometric deep learning	http://bioinfo3d.cs.tau.ac.il/ScanNet/index_real.html (accessed on 15 August 2024)	[162]

NA: not available.

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
