# Peer review of "Allergies to Allergens from Cats and Dogs: A Review and Update on Sources, Pathogenesis, and Strategies"

_ijms, 2024, doi:10.3390/ijms251910520_

Round 1

Reviewer 1 Report

Comments and Suggestions for Authors

Comments: Allergies to Allergens from Cats and Dogs: A Review and Up-date on Sources, Pathogenesis, and Strategies

This review provides comprehensive insights into the classification of cat and dog allergens, along with their pathogenic mechanisms and current therapeutic strategies, which makes the review of interest for the topic and relevant for the readers of this journal.

1. The authors should summarize existing problems about current strategies for preventing and treating allergies.

2. In abstract, “This review will improve knowledge regarding allergies to cats and dogs, while identifying potential targets for the development of next-generation treatments”. However, this review does not involve in the identification of potential targets for the development of next-generation treatments.

3. There are few typos in the manuscripts kindly give a thorough revision before the final submission. Such as Line 21 “…comprhensive…cat and dog allerges…”

4. The Figures and Tables are not showed in the manuscript, which makes commenting of the manuscript difficult.

Comments on the Quality of English Language

Moderate editing of English language required.

Author Response

请参阅附件

Reviewer 2 Report

Comments and Suggestions for Authors

Peer review of the article: “Allergies to Allergens from Cats and Dogs: A Review and Up- 2

date on Sources, Pathogenesis, and Strategies”.

This review provides a comprehensive insight into the classification of cat and dog allergies, along with their pathogenic mechanisms. The study also discusses implementation strategies for prevention and control measures, including physical methods, gene editing technology, immunological approaches, as well as potential strategies for enhancing allergen immunotherapy combined with immunoinformatic.

It also presents prospects for the prevention and treatment of human allergies caused by cats and dogs. This review will improve knowledge regarding allergies to cats and dogs, while identifying potential targets for the development of next-generation treatments.

General Information provides highlights that the inhalation of cat and dog allergens by humans can lead to a range of discomforting symptoms, such as rhinitis and asthma. Given the increasing popularity of these animals, the allergens they produce pose a growing threat to the health of susceptible individuals. Allergens from cats and dogs have emerged as significant risk factors for allergic sensitization and triggering asthma and allergic rhinitis worldwide. However, there remains a lack of systematic measures aimed at assisting individuals in recognizing and preventing allergies caused by these animals.

Specific Comments:

The article contains the most relevant information to the subject. However this reviewer feels that the paper could be slightly shortened and the number of references reduced.

The English language should be revised and slight simplified and avoid some repetitions. The authors should provide a clear message on what has been done and should be further pursued in this important allergic syndrome. Pet allergies could also pose a problem in schools and public places since their allergens are easily transported on clothing.

Interesting subject which contains most of the most relevant information.

Comments on the Quality of English Language

Peer review of the article: “Allergies to Allergens from Cats and Dogs: A Review and Up- 2

date on Sources, Pathogenesis, and Strategies”.

This review provides a comprehensive insight into the classification of cat and dog allergies, along with their pathogenic mechanisms. The study also discusses implementation strategies for prevention and control measures, including physical methods, gene editing technology, immunological approaches, as well as potential strategies for enhancing allergen immunotherapy combined with immunoinformatic.

It also presents prospects for the prevention and treatment of human allergies caused by cats and dogs. This review will improve knowledge regarding allergies to cats and dogs, while identifying potential targets for the development of next-generation treatments.

General Information provides highlights that the inhalation of cat and dog allergens by humans can lead to a range of discomforting symptoms, such as rhinitis and asthma. Given the increasing popularity of these animals, the allergens they produce pose a growing threat to the health of susceptible individuals. Allergens from cats and dogs have emerged as significant risk factors for allergic sensitization and triggering asthma and allergic rhinitis worldwide. However, there remains a lack of systematic measures aimed at assisting individuals in recognizing and preventing allergies caused by these animals.

Specific Comments:

The article contains the most relevant information to the subject. However this reviewer feels that the paper could be slightly shortened and the number of references reduced.

The English language should be revised and slight simplified and avoid some repetitions. The authors should provide a clear message on what has been done and should be further pursued in this important allergic syndrome. Pet allergies could also pose a problem in schools and public places since their allergens are easily transported on clothing.

Interesting subject which contains most of the most relevant information.

Author Response

请参阅附件。

Reviewer 3 Report

Comments and Suggestions for Authors

In their review the authors comprehensively present vital aspects of cat and dog allergy as well as highlight crucial issues regarding development of prospective treatment modalities.

In my opinion the article is well-structured and easy to navigate, logically presenting the above mentioned topics.

The bibliography is properly selected and covers most of the important sources that have appeared with the reviewed field over last period. Regarding references, however, I have noticed that the Authors do not refer to the Molecular Allergology User’s Guide 2.0 that has been issued recently by the Euopean Academy of Allergy and Clinical Immunology (EAACI) and published in the “Pediatric Allergy and Immunology” journal (Pediatr Allergy Immunol. 2023 Mar:34 Suppl 28:e13854. doi: 10.1111/pai.13854). This is quite an important collection of current concepts and knowledge about molecular allergology both in terms of allergens’ structure and significance of their assessment in the diagnostic process. As such, I believe it deserves to be referenced in this kind of publication.

Another major issue is that there are several tables mentioned in the text, namely Table 1, Table 2 and Table 4. However, they are not included in the manuscript text nor was there any downloadable supplementary material. In addition, table 3 is missing. It is necessary to clarify if the table are finally meant to be included in the manuscript and if yes, to include them in the revised version.

Minor issues:

Lines 35-36: please consider providing some more specific data to replace the statement “prevalent global condition”. Besides, referring allergy in total here (as far as I understood) seems to general – data on the prevalence of exemplary allergic disease would look more clear.

Line 47: I suggest clarifying whether 1/5 refers to the general population and if this prevalence data refers to the symptomatic or asymptomatic sensitization, or both.

Lines 163-170: The Authors may considering creating a separate sub-chapter to discuss the Can f 5 allergen, which is clinically interesting and  important and determining sensitivity solely to male dogs.

Author Response

请参阅附件。

Round 2

Reviewer 1 Report

Comments and Suggestions for Authors

No further conments 

Reviewer 3 Report

Comments and Suggestions for Authors

The Authors have adequately responded to my comments and queries.

Please find below some minor remarks regarding the revised manuscript

1. Table 1 - I suggest modifying the title to: "Frequency of sensitization to cat and dog allergens in different countries" to better reflect the contents.

2. In amended lines 175-185, regarding Can f 5 allergen molecule, the sentence: "There have been reported cases where individuals who previously had no adverse reactions to semen exposure developed allergies to semen in public places". The last fragment "allergies to semen in public places" - I suggest modifying it into: ''(...) allergic symptoms upon exposure to dogs in public places". If my suggestion is not what the Authors had in mind, then keep it as it is.

3. Table 2 - please add the horizontal line to divide the part describing cat and dog allergens. Moreover, I suggest modifying the title to: Allergenic components of cat and dog allergens" , or similar.

Otherwise, no issue to be raised. Thanks again for the possibility to review this manuscript.
